# Tunable Ag Nanocavity Enhanced Green Electroluminescence from SiN_x_:O Light-Emitting Diode

**DOI:** 10.3390/nano14151306

**Published:** 2024-08-03

**Authors:** Zongyan Zuo, Zhongyuan Ma, Tong Chen, Wenping Zhang, Wei Li, Jun Xu, Ling Xu, Kunji Chen

**Affiliations:** 1School of Electronic Science and Engineering, Nanjing University, Nanjing 210093, China; 2Collaborative Innovation Center of Advanced Microstructures, Nanjing University, Nanjing 210093, China; 3Jiangsu Provincial Key Laboratory of Photonic and Electronic Materials Sciences and Technology, Nanjing University, Nanjing 210093, China

**Keywords:** localized surface plasmon, nanocavity, SiN_x_, electroluminescence

## Abstract

As the driving source, highly efficient silicon-based light emission is urgently needed for the realization of optoelectronic integrated chips. Here, we report that enhanced green electroluminescence (EL) can be obtained from oxygen-doped silicon nitride (SiN_x_:O) films based on an ordered and tunable Ag nanocavity array with a high density by nanosphere lithography and laser irradiation. Compared with that of a pure SiN_x_O device, the green electroluminescence (EL) from the SiN_x_:O/Ag nanocavity array device can be increased by 7.1-fold. Moreover, the external quantum efficiency of the green electroluminescence (EL) is enhanced 3-fold for SiN_x_:O/Ag nanocavity arrays with diameters of 300 nm. The analysis of absorption spectra and the FDTD calculation reveal that the localized surface plasmon (LSP) resonance of size-controllable Ag nanocavity arrays and SiN_x_:O films play a key role in the strong green EL. Our discovery demonstrates that SiN_x_:O films coupled with tunable Ag nanocavity arrays are promising for silicon-based light-emitting diode devices of the AI period in the future.

## 1. Introduction

With the era of AI and big data coming, silicon-based optoelectronic integration is critical to constructing optic I/O interfaces for the next generation of GPUs with high speeds [1,2,3,4,5]. Compared with electrons, photons have the advantages of low transmission loss and fast transmission speed. As the driving source, silicon-based light emission paves the way for optoelectronic integration chips. Currently, there are three main silicon-based light sources, including external light sources, on-chip integrated III-V compound semiconductor epitaxy on silicon substrates and silicon-based luminescent materials [6,7,8]. External light sources are not conducive to large-scale on-chip integration, and on-chip integrated III-V compound semiconductor epitaxy on silicon substrates faces lattice mismatch and CMOS process incompatibility. Therefore, highly efficient silicon-based light-emitting materials are critical for building on-chip optical interconnection because of their compatibility with the CMOS technology [9,10,11,12,13,14]. Among the expanded species of silicon-based light emission materials [15,16,17,18], amorphous oxygen-doped silicon nitride (SiN_x_:O) films are attractive candidates due to their tunable band gap and low fabrication temperature [19,20]. The external quantum efficiency of pure SiN_x_:O is comparable to that of traditional fluorescent organic light-emitting diodes (OLEDs). Due to the contribution of the singlet excitons, the theoretical maximum external quantum efficiency of these OLEDs is about 5–7.5% [21]. Until now, the external quantum efficiency of SiN_x_:O-based LEDs was lower than that of the aggregation-induced delayed fluorescence (AIDF) OLEDs (over 20%) [21,22]. To allow the Si-based emission device with strong electroluminescence (EL) to be used in optoelectronic integration [23,24], enhanced spontaneous emission has been achieved by using the coupling of emitters and localized surface plasmon (LSP) of the metal cavity [25,26,27,28,29]. Since the resonance effect of the metal cavity can increase the radiation recombination rate of the emitter [30,31,32,33], the LSP of metal particles was adopted to increase the luminescence efficiency of SiN_x_:O [34]. However, the enhancement of luminescence efficiency is lower due to the random size of metal particles. [35,36,37,38,39]. Using hexagonal arrays of size-controllable Ag nanoparticles fabricated by nanosphere lithography, the LSP resonance can effectively enhance the luminescence efficiency of SiN_x_:O films [40]. But the density of Ag nanoparticles is lower as the surface coverage is 14.4%. So, the luminescence enhancement is limited. To further enhance the electroluminescent efficiency of the SiN_x_:O emitter, size-tunable metal nanocavity arrays with high density are urgently needed.

Here, we are committed to improving the electroluminescent efficiency of SiN_x_:O films by using size-tunable Ag nanocavity arrays with high density through nanosphere lithography and laser irradiation. The surface coverage of the Ag nanocavity arrays reaches 78.5%. The surface of polystyrene (PS) nanosphere arrays as the template was covered with thin Ag films by sputtering technology. Ag-PS core–shell arrays were irradiated by the pulsed laser to form the orderly Ag nanocavity arrays. The LSP resonance wavelength of the Ag nanocavity arrays can be tuned by changing the size of the Ag nanocavity. LSP resonance wavelength redshifts as the diameter of the Ag nanocavity increases from 170 to 300 nm. Under the effective coupling of the Ag nanocavity arrays, 7.1-fold EL enhancement was acquired in the system of SiN_x_:O/Ag nanocavity arrays. Furthermore, the external quantum efficiency of the SiN_x_:O/Ag nanocavity arrays is increased by 3-fold compared with that of pure SiN_x_:O films. Our discovery demonstrates that SiN_x_:O films coupled with tunable Ag nanocavity arrays are promising for silicon-based light-emitting diode devices of the AI period in the future.

## 2. Materials and Methods

The fabrication process diagram of the SiN_x_:O/Ag nanocavity arrays is displayed in Figure 1a. First, a p-type silicon wafer was employed as the substrate. PS spheres with different diameters are dropped onto silicon wafers and then transferred to deionized water. PS spheres were produced by the company Macklin (Shanghai, China). Its weight volume is 5%. Due to the self-assembly characteristics of PS spheres, a transparent single-layer array of PS spheres will float on the surface of deionized water. To obtain a film that has a large number of PS spheres, repeat the steps mentioned above multiple times. Use a cleaned silicon substrate to remove the PS spheres film in deionized water with a bottom–top transfer strategy, ensure that the surface of the silicon substrate is covered with PS spheres, and then tilt the substrate on the filter paper to air dry naturally. The magnetron sputtering system was used to coat the Ag films on the surface of the PS spherical arrays. The purity of the Ag target is 99.99%, which was produced by the company ZhongNuo Advanced Material (Beijing, China). The vacuum of magnetron sputtering system is 10^−4^ Pa. The sputtering power is 50 W. The pulsed laser (with a frequency of 50 Hz and pulse width of 20 ns; COMPex 205, Coherent, PA, USA) with a wavelength of 351 nm was used to irradiate the Ag-PS spherical arrays. The instantaneous high temperature caused by laser irradiation made PS spheres melt or even evaporate, and Ag condensed into new nanocavities. Thus, the silicon substrate was covered with a new Ag nanocavity array. The SiN_x_ films were deposited onto the Ag nanocavity arrays by PECVD. The flow rates of silane (SiH_4_) and ammonia (NH_3_) were maintained at 8 sccm and 48 sccm, respectively. The SiN_x_ films with a thickness of 350 nm were deposited on the Ag nanocavity arrays with different diameters, which guaranteed that the Ag nanocavity arrays were completely encased in the SiN_x_ films. Finally, in situ oxidation of SiN_x_ films was performed for 30 min, ensuring the formation of SiN_x_:O films. To ensure the successful EL measurement of the SiN_x_:O/Ag nanocavity arrays device, the top Al electrodes were fabricated by the electron beam deposition using a hard mask with circle holes. And the bottom electrodes were obtained by deposition of Al on the backside of the substrate. The EL measurement schematic diagram of Al/SiN_x_:O/Ag nanocavity arrays/p-Si/Al device is shown in Figure 1b.

The absorption spectra were measured by the Shimadzu UV-3600 spectrophotometer (Shimadzu, Kyoto, Japan). HORIBA JOBIN YVON fluor ESSENCE spectrofluorometer (HORIBA, Kyoto, Japan) was used to measure the photoluminescence (PL) spectra. The photoluminescence quantum yield (PLQY) was measured using an integration sphere with a xenon lamp as the excitation light source. The time-resolved PL (TRPL) decay spectrum was measured to obtain the lifetime of carriers by using TCSPC. The Agilent B1500A semiconductor analyzer (Agilent, CA, USA) was used to investigate the electrical behavior of the devices. The external quantum efficiency is obtained from the current density–luminescence curves and EL spectra. The current density–luminescence curves and EL spectra were measured with a combination of the Keithley 2400 source meter (Keithley, Shanghai, China) and an integrating sphere (FOIS-1, Ocean Optics, Shanghai, China) coupled with QE65 Pro spectrometer (Ocean Optics, Shanghai, China). The continuous laser (UV-FN-360, CNIlaser, Jilin, China) with a corresponding wavelength of 360 nm was used to measure the PL spectra of SiN_x_:O films with and without Ag nanocavity arrays. The LSP electric field intensity distribution of Ag nanocavity arrays was calculated using the finite difference time domain (FDTD) model. The plane wave was used as the light source in the FDTD. It was located vertically above the substrate and covered the Ag nanocavity arrays entirely. The perfectly matched layer (PML) was chosen as the boundary condition to avoid the reflection effect of electromagnetic waves.

## 3. Results and Discussion

The SEM and AFM images of the Ag-coated PS spherical arrays are displayed in Figure 2. The PS spheres with three different diameters, including 170 nm, 220 nm and 300 nm, were used as the template to fabricate the Ag-PS core–shell structure. Figure 2a exhibits the cross-section SEM images of Ag-PS core–shell arrays with three different diameters. Figure 2b shows the top AFM images of the Ag-PS core–shell arrays with three different diameters. As shown in Figure 2c, the diameter of 170 ± 2nm is estimated from the SEM images when the template is the PS spherical arrays with a sphere diameter of 170 nm. The diameter of 220 ± 5 and 300 ± 3 nm of Ag-PS core–shell nanocavities correspond to the PS sphere with a diameter of 220 and 300 nm, respectively. In all three samples, the periodic order was demonstrated. Note that the surface coverage of the Ag nanocavity arrays reaches 78.5%.

In order to investigate the coupling effect between LSP of the Ag nanocavity arrays and SiN_x_:O films, the absorption spectra and the PL spectra of pure SiN_x_:O films and the SiN_x_:O/Ag nanocavity arrays were measured as displayed in Figure 3a. It is observed that two peaks coexist in the whole absorption spectra from the three kinds of devices with different Ag nanocavity diameters. One is located at 422 nm, and the other one is centered at 489 nm when the diameter of the Ag nanocavity is 170 nm. As for the Ag nanocavity with a diameter of 220 nm and 300 nm, the absorption peak at 489 nm moves to 496 nm and 508 nm, respectively. The redshift of the absorption peak is due to the increase in Ag nanocavity diameter, implying that controlling the diameter of the PS sphere can modify the resonance wavelength of the Ag nanocavity arrays. It is obvious that the position and intensity of the absorption peak at 422 nm remain unchanged, with the diameter increasing from 170 to 300 nm. As shown in Figure 3a, the absorption peak of pure SiN_x_:O films is also located at 422 nm, which is in accordance with that of the fixed absorption peak from the spectra of SiN_x_:O/Ag nanocavity arrays. The unchanged position and intensity of the absorption peak at 422 nm further reveal that the amount of SiN_x_:O on the surface of the three different Ag nanocavity arrays is equal to each other. The pump source with the wavelength of 360 nm was used to measure the PL spectra of pure SiN_x_:O films and SiN_x_:O/Ag nanocavity arrays. The PL signal of pure SiN_x_:O films is a wide emission spectrum with a peak of 523 nm, as shown in Figure 3b. The PL emission originates from the radiative recombination of electrons and holes in luminescent centers related to the Si–O bonds, as described in our previous work [19]. 

After the introduction of the Ag nanocavity arrays, the PL peak moves to a new band with a shorter wavelength. When the diameter of the Ag nanocavity decreases from 300 nm to 170 nm, the PL peak blue shifts from 515 nm to 496 nm. It is evident that the PL peak can also be changed by the Ag nanocavity arrays with different diameters, which display the blue shift trend the same as that of the corresponding absorption spectra. It is worth noting that the PL peak wavelength is close to the resonance peak of the absorption spectra. The overlapping of the PL peak and absorption peak clearly indicates that the PL emission behavior is regulated by the LSP of the Ag nanocavity arrays. In addition, the photoluminescence quantum yield (PLQY) of SiN_x_:O films with and without Ag nanocavity arrays were systematically measured. The PLQY values are 38.6% for pure SiN_x_:O films, 42.6% for SiN_x_:O/Ag nanocavity arrays with a diameter of 170 nm, 46.3% for SiN_x_:O/Ag nanocavity arrays with a diameter of 220 nm, and 58.2% for SiN_x_:O/Ag nanocavity arrays with the diameter of 300 nm, respectively. At the same time, the fluorescence lifetime (τ) of these devices was also measured. The lifetime values of 11.8 ns, 8.3 ns, 7.1 ns and 4.3 ns belong to pure SiN_x_:O films, with the SiN_x_:O/Ag nanocavity arrays with the Ag nanocavity diameter increasing from 170 nm to 300 nm. As can be seen, the PLQY increases, and the lifetime decreases with the increase in the diameter of the Ag nanocavity arrays. According to the PLQY and lifetime values, we calculated the radiative decay rate (k_r_ = PLQY/τ) of SiN_x_:O films with and without Ag nanocavity arrays. The k_r_ value of 3.27 × 10^7^ s^−1^ was obtained in pure SiN_x_:O films. The k_r_ values of SiN_x_:O/Ag nanocavity arrays are 5.13 × 10^7^ s^−1^, 6.52 × 10^7^ s^−1^ and 1.35 × 10^8^ s^−1^ as the Ag nanocavity diameter increases from 170 nm to 300 nm. The increase in k_r_ value indicates that the spontaneous emission of SiN_x_:O films has been enhanced with the diameter of the Ag nanocavity arrays increasing, which is related to the coupling between the SiN_x_:O films and the Ag nanocavity arrays. The larger the k_r_ value, the stronger the coupling strength and the PL enhancement are. Therefore, the PL emission intensity of SiN_x_:O films has been greatly improved with the coupling of the Ag nanocavity arrays. The PL enhancement factor is the ratio of the integral PL spectrum of SiN_x_:O/Ag nanocavity arrays with different diameters and pure SiN_x_:O films. When the diameter of the Ag nanocavity is 170 nm, a 1.3-fold PL enhancement factor is obtained from the PL spectra of SiN_x_:O/Ag nanocavity arrays. A 2.1-fold PL enhancement factor is acquired in the SiN_x_:O/Ag nanocavity arrays when the diameter of the Ag nanocavity is 220 nm. When the diameter of the Ag nanocavity is 330 nm, a 4.8-fold PL enhancement factor is detected, which is related to the largest overlap between the absorption peak of the SiN_x_:O/Ag nanocavity arrays and the PL peak of the pure SiN_x_:O films.

The photographs of the EL devices based on the SiN_x_:O/Ag nanocavity arrays with various diameters are shown in Figure 4a. The inset shows the image of the green EL emission in a darker environment from metal-free central area rounded by the cathode. The surface of the SiN_x_:O/Ag nanocavity arrays displays different colors, with the diameter of the Ag nanocavity changing from 170 to 300 nm, which is induced by the various LSP resonant wavelengths of the Ag nanocavity arrays. As reported in reference [41], different materials and sidewall emissions cause the mismatched angular distribution of the emission, which is the reason for the color shift. In our work, different materials and sidewall emissions are not present. It is only induced by the LSP resonant wavelength of the Ag nanocavity arrays with various diameters. Note that the wavelength of LSP mode does not vary with the angle. Therefore, our EL devices have the same angular distribution of the emission. Figure 4b shows the current–voltage (I-V) characteristics of the pure SiN_x_:O films and SiN_x_:O/Ag nanocavity arrays. The threshold voltage of the SiN_x_:O/Ag nanocavity array devices is lower than that of the pure SiN_x_:O films. The lower threshold voltage reveals that the Ag nanocavity arrays can significantly enhance the carrier injection into the SiN_x_:O films. 

Figure 5 shows the EL spectra of the SiN_x_:O/Ag nanocavity arrays and pure SiN_x_:O films as a function of the injection voltage. As for the pure SiN_x_:O films, the main peak of the EL spectra is located at 550 nm, as displayed in Figure 5a. The EL intensity increases with the bias changing from 6 to 12 V. After the introduction of the Ag nanocavity arrays with a diameter of 170 nm, the EL peak blueshifts to 480 nm, as presented in Figure 5b. As shown in Figure 3a, an obvious peak at 489 nm is detected in the absorption spectrum when the diameter of the Ag nanocavity is 170 nm, which is closer to the peak wavelength of EL spectra. It reflects the coupling effect of LSP and SiN_x_:O films when the EL spectra and the absorption spectra overlap. The EL intensity is increased compared with that of the pure SiN_x:_O films under the voltage scope from 6 to 12 V. The EL enhancement factor is 1.5-fold, which is defined as the ratio of the integral EL intensity of SiN_x_:O/Ag nanocavity and pure SiN_x_:O films. It is interesting to find that the EL peak redshifts to 520 nm when the diameter of the Ag nanocavity is enhanced to 220 nm, as shown in Figure 5c. The EL intensity increases more strongly than that of the device with an Ag nanocavity diameter of 170 nm when the voltage ranges from 6 to 12 V. And a 2.3-fold EL enhancement factor is acquired from the EL devices. When the diameter of the Ag nanocavity is enhanced to 300 nm, the EL peak redshifts further to 535 nm, as shown in Figure 5d. The EL enhancement factor increases sharply, reaching a maximum of 7.1-fold. The redshift of the EL peak with the increase in the diameter of the Ag nanocavity is also due to the coupling effect of LSP and SiN_x_:O films. Because the absorption peak redshifts with the increase in the diameter of Ag nanocavity, which is in accordance with the report of reference [42]. The increase in diameter causes the LSP resonance peak to become closer to the EL peak of pure SiN_x_:O films, leading to the EL peak of SiN_x_:O/Ag nanocavity arrays to redshift. Considering that the EL peak of 535 nm from the SiN_x_:O/Ag nanocavity arrays with a diameter of 300 nm is closest to that of pure SiN_x_:O films, the coupling strength of SiN_x_:O films and Ag nanocavity arrays is strongest. The maximum EL enhancement factor is in agreement with that of PL enhancement, as presented in Figure 3b. 

Figure 6a displays the energy band diagram of SiN_x_:O/Ag nanocavity/p-Si under zero bias. The band gap (E_g_) of SiN_x_:O films and p-Si is 4.5 eV and 1.12 eV, respectively. Under the forward bias, the holes in p-Si migrate to the luminescent centers related to Si–O in the SiN_x_:O films to complete the radiative recombination with the electrons, yielding the green emission, as shown in Figure 6b. The resonance coupling of these emitted photons and the LSP mode results in an enhancement of EL. The external quantum efficiency (EQE) is an important index for evaluating the performance of LED devices, which is the ratio of the number of photons emitted into space per unit of time and the number of electrons injected into the active layer per unit of time. Figure 6c shows the EQE of pure SiN_x_:O films and SiN_x_:O/Ag nanocavity arrays as a function of current density. The EQE of pure SiN_x_:O films and SiN_x_:O/Ag nanocavity arrays with different diameters were all achieved at the current density of 92 mA/cm^2^. The EQE of SiN_x_:O/Ag nanocavity arrays with the diameter of 170, 220 and 300 nm is 3.6%, 3.9% and 9.5% and is enhanced by 1.1-, 1.2- and 3-fold compared with that of pure SiN_x_:O films. 

To find further insights into the coupling effect between SiN_x_:O films and the LSP mode of the Ag nanocavity arrays, we used FDTD to calculate the electric field intensity distribution of the Ag nanocavity arrays embedded in SiN_x_:O films, as shown in Figure 7. The increased electric field intensity can be observed from the x-y plane and the x-z plane, with the diameter of the Ag nanocavity increasing from 170 to 300 nm, which confirms the near-field enhancement characteristic of LSP between the Ag nanocavity arrays and SiN_x_:O matrix. 

## 4. Conclusions

In summary, enhanced green EL is successfully obtained from the SiN_x_:O/Ag nanocavity arrays with a high density by nanosphere lithography and laser irradiation. The absorption peak redshifted from 475 to 500 nm as the diameter of the Ag nanocavity increases from 170 to 300 nm. When the diameter of the Ag nanocavity is 300 nm, the resonance peak induced by the LSP of the Ag nanocavity arrays has the largest overlap with the EL spectra of pure SiN_x_:O films. The analysis of absorption spectra and the FDTD calculation reveal that the LSP resonance of size-controllable Ag nanocavity arrays can effectively enhance the green EL from SiN_x_:O films by 7.1-fold. Moreover, the external quantum efficiency of the green EL device can be increased by 3-fold. Our discovery demonstrates that SiN_x_:O films coupled with tunable Ag nanocavity arrays are promising for silicon-based light-emitting diode devices of the AI period in the future.

## Figures and Tables

**Figure 1 nanomaterials-14-01306-f001:**
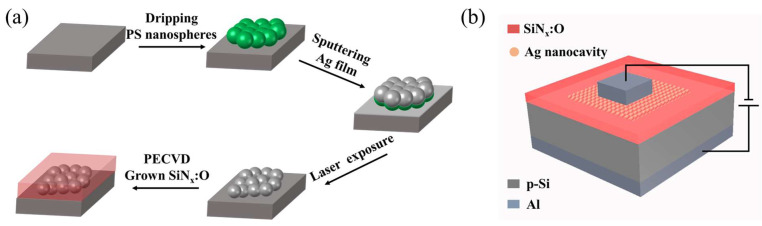
(**a**) The fabrication process diagram of SiN_x_:O/Ag nanocavity arrays device. (**b**) The EL measurement schematic diagram of the SiN_x_:O/Ag nanocavity arrays device.

**Figure 2 nanomaterials-14-01306-f002:**
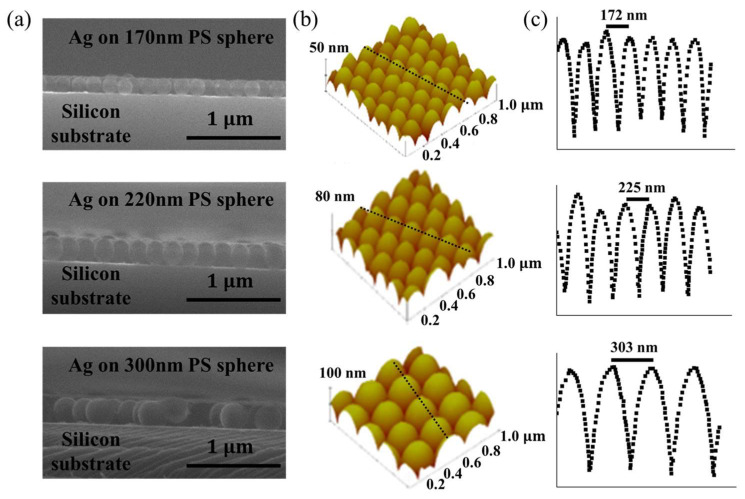
(**a**) The SEM images of the Ag-PS core–shell spherical arrays with three different diameters. The diameter of PS sphere is 170 nm, 220 nm and 300 nm, respectively. (**b**) The AFM images of the Ag-PS core–shell spherical arrays with three different diameters. (**c**) The cross-section profile of dotted line position of the AFM images corresponds to the Ag-PS core–shell spherical arrays with three different diameters.

**Figure 3 nanomaterials-14-01306-f003:**
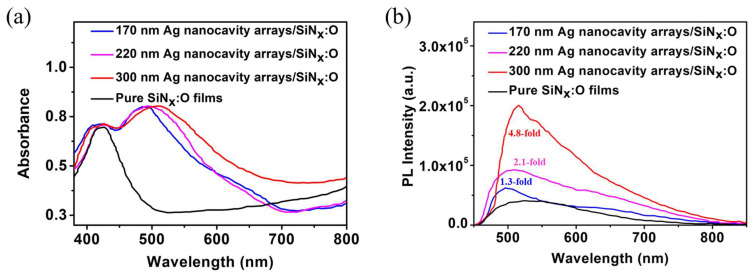
(**a**) The absorption spectra of pure SiN_x_:O films and the SiN_x_:O/Ag nanocavity arrays. (**b**) The PL spectra of pure SiN_x_:O films and the SiN_x_:O/Ag nanocavity arrays.

**Figure 4 nanomaterials-14-01306-f004:**
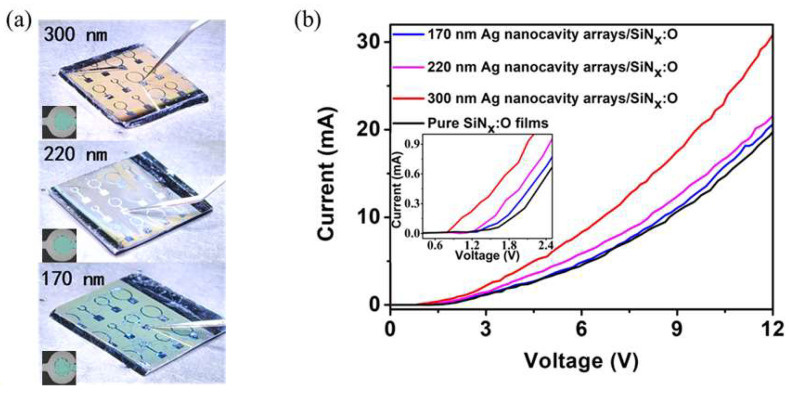
(**a**) The photographs of the EL device based on the SiN_x_:O/Ag nanocavity arrays. Inset shows the image of the emission from metal-free central area rounded by the cathode. (**b**) Current–voltage characteristics of the pure SiN_x_:O films and SiN_x_:O/Ag nanocavity array devices.

**Figure 5 nanomaterials-14-01306-f005:**
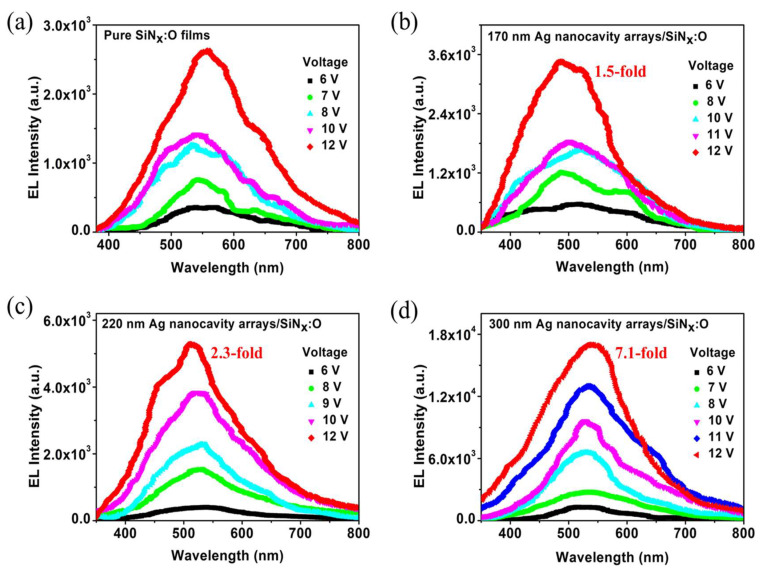
(**a**) The EL spectra of the pure SiN_x_:O films. The EL spectra of SiN_x_:O/Ag nanocavity array devices when the diameter of Ag nanocavity is (**b**) 170 nm, (**c**) 220 nm and (**d**) 300 nm, respectively.

**Figure 6 nanomaterials-14-01306-f006:**
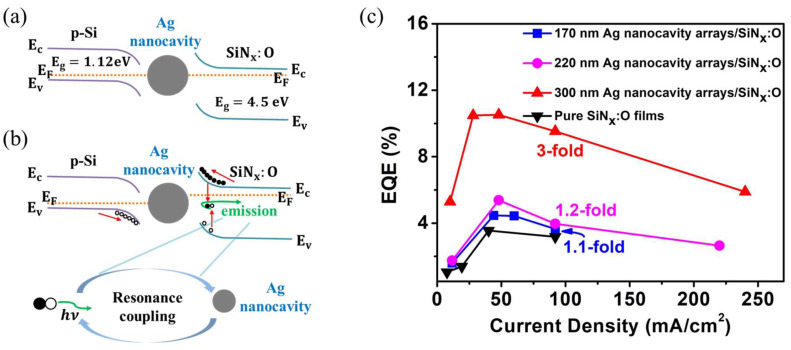
(**a**) The band diagram of SiN_x_:O/Ag nanocavity/p-Si device under zero bias. (**b**) The band diagram of SiN_x_:O/Ag nanocavity/p-Si device under forward bias. (**c**) The EQE of SiN_x_:O films and SiN_x_:O/Ag nanocavity as a function of current density.

**Figure 7 nanomaterials-14-01306-f007:**
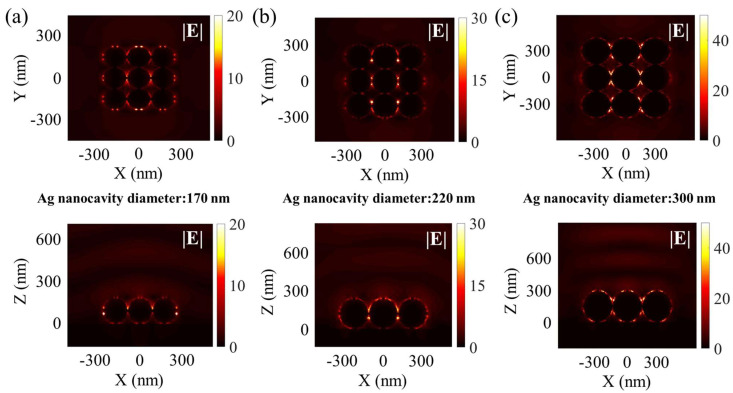
The electric field distribution of the SiN_x_:O/Ag nanocavity arrays with different diameters viewed from the x-y plane and the x-z plane, which corresponds to the resonance mode of the Ag nanocavity arrays. The diameter of the Ag nanocavity is (**a**) 170 nm, (**b**) 220 nm and (**c**) 300 nm, respectively. The simulated wavelengths correspond to that of the experimental EL peaks from the SiN_x_:O/Ag nanocavity arrays.

## Data Availability

Data are contained within the article.

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
