# Peer review of "Tunable Ag Nanocavity Enhanced Green Electroluminescence from SiNx:O Light-Emitting Diode"

_nanomaterials, 2024, doi:10.3390/nano14151306_

Round 1

Reviewer 1 Report (Previous Reviewer 2)

Comments and Suggestions for Authors

All of the queries were solved. This manuscript is now suitable for publication.

Author Response

Thanks for the great help of the referee.

Reviewer 2 Report (New Reviewer)

Comments and Suggestions for Authors

The manuscript demonstrates a significant enhancement in green electroluminescence (EL) from oxygen-doped silicon nitride (SiNx:O) films using Ag nanocavities. This results in a 7.1-fold increase in EL and a 3-fold improvement in quantum efficiency. The findings suggest promising applications for silicon-based LEDs. The manuscript is well written and the data is well organized. It is my pleasure to recommend this work for publication. The reviewer has just some comments to the manuscript.

1.       It would be beneficial to compare the EQE of green OLEDs with that of silicon-based green LEDs. Additionally, please include recent references related to OLEDs emitting at green wavelengths in the revised manuscript, such as Yang et al. (https://doi.org/10.1016/B978-0-12-824335-0.00016-7).

2.      The author needs to explain the type and mechanism of fluorescence and electroluminescence (EL) involved.

3.      It is important to provide the photoluminescence quantum yield for each of the emitters.

4.      I recommend calculating and providing the kinetic parameters, such as rate constant to enhance the clarity of fluorescence mechanism for each of the emitters.

Comments on the Quality of English Language

The manuscript is mostly fine, but some areas need better phrasing and grammar.

Author Response

Reviewer 3 Report (New Reviewer)

Comments and Suggestions for Authors

In this manuscript “Tunable Ag nanocavity enhanced green electroluminescence from SiNx:O light emitting diode,” the authors demonstrated an interesting approach by using tunable Ag nanocavities to enhance the EL intensity by 7.1x and EQE by 3x. The content is well-structured, with sufficient characterization details, and offers a comprehensive discussion, and more importantly, the results are impressive. I recommend it for publication after minor revisions. Some specific questions and comments are listed as follows.

1.     It is difficult to see the emitting differences between the devices shown in Figure 4a. Could the authors take the pictures in a darker environment?

2.     Typically, nanocavity in emitting devices will also impact the angular emission distribution, which in turn causes color shift, as analyzed and observed by F. Gou, et al. Opt. Express 27(12), A746-A756 (2019). Could you provide more details on whether the angular distribution is changed in this case?

Author Response

This manuscript is a resubmission of an earlier submission. The following is a list of the peer review reports and author responses from that submission.

Round 1

Reviewer 1 Report

Comments and Suggestions for Authors

The article "Tunable Ag nanocavity enhanced green electroluminescence 2 from SiNx:O light-emitting diode"  demonstrates that SiNx:O films coupled with tunable Ag nanocavity arrays are promising for silicon-based light-emitting diode devices. The obtained results are very interesting for the readers of  Nanomaterials journal.

Before the final decision some additional data must be presented, as well as some question must be clear up.

The polystyrene material must be described in details (Producer, purity quality. ets.)/ The same to Ag tagret for magnetron sputtering.

Magnetron sputtering parameters must be presented/

Laser details (model, wavelength, repetition rates, Producer, etc.) must be presented.

"The extinction spectra.."' change to "The absorption spectra..."

In the Introduction section the authors wrote "the external quantum efficiency of the SiNx:O/Ag nanocavity arrays is increased 50 by 3-fold compared with that of pure SiNx:O films." But there were not presented any details about quantum yield measurements.

In Figure 7 the scale mark must be presented.

Reviewer 2 Report

Comments and Suggestions for Authors

                  This manuscript reports electroluminescence of oxygen-doped silicon nitride (SiNx:O) films wth Ag nanocavities formed by nanosphere lithography and laser irradiation. This result could be helpful for the development of silicon-based light source for optic I/O interface. However, I have some issues with the manuscript as written that must be addressed before it is suitable for publication in Nanomaterials. I recommend a publication after major revision with additional data and a better structuring of the findings.

1. Authors should revised their Abstract. The first sentence should summarize their findings not introduction.

2. Authors stated in line 22 that “promising for silicon-based light-emitting diode devices of AI period in the future.” Green LED is already well developed and shows strong enough intensity; therefore, it is necessary to explain exactly how helpful this study is and what this has to do with the AI era.

3. In order to compare the PL intensities, first under the assumption of the sample SiNx:O thickness is thin enough to satisfy absorption and PL intensity show linear relationship, standard absorption spectra are required. The normalized extinction shown in Figure 3(a) is not helpful for PL intensity comparison.

4. The EL intensity spectra from microsphere as shown in Figure 5 can’t be a proof for enhancement of EL by localized surface plasmon. A wide diameter of sphere means that the valley between the peaks is deep, resulting in more oxygen-doped silicon nitride existence. Therefore it is not correct for the EL intensity comparison of SiNx:O film with different diameter sphere.

5. Author should explain the reason for blue shift of EL emission peak position at external bias of 12V as shown in Figure 5. (i.e. pure SiN:O : ~ 550 nm, SiN:O on 170 nm cavity : ~ 480 nm). Author should discuss more for the reason of red shift as the diameter of sphere increases.

6. The most important thing in LED study is a quantum efficiency. What is the reason for not reporting the quantum efficiency plot? Authors stated that “electroluminescent internal efficiency of SiNx:O” (in line 41) and the external quantum efficiency of the SiNx:O/Ag nanocavity arrays is increased by 3-fold compared with that of pure SiNx:O films” (in line 50 ~ 51), there is no quantum efficiency plot. There is a normalized EL intensity vs current plot (Figure 6(c)) in the manuscript. Authors should obtain the quantum efficiency and insert it within the manuscript.

7. The references are too old. LED fields are improving day by day. It is not right to reference these ole reports as a reference.

8. First of all, the biggest problem of this study is lack of originality. What is the originality of this study compare to already reported luminescence efficiency enhancement of SiNx:O films by surface plasmon effect of a convex structure. Authors should explain what is the differences and originality of their study.

Comments on the Quality of English Language

Minor editing is required.

Round 2

Reviewer 2 Report

Comments and Suggestions for Authors

             The authors misunderstood my points about PL and EL intensity. As I pointed out in my first reviewer’s comments (Q3 was related to PL intensities for Figure 3(a) and Q4 was related to EL intensities for Figure 5), PL and EL intensities are closely related to the amount of chromophore. Of course, as the authors responded, there should be a contribution from the coupling effect induced by the localized surface plasmon for these PL and EL intensities. However, a contribution from the differences in SiNx:O chromophore amount should also be counted. That is why I asked for standard absorption spectra to prove equal amounts of SiNx:O chromophore were used to compare PL and EL intensities. If the absorbance of SiNx:O chromophore with different sizes of nanocavity, the authors should make a correction for the PL and EL intensities. Unless the authors prove this point (i.e. use of equal amounts of SiNx:O chromophore so that there is only the localized surface plasmon effect for these PL and EL intensities), the conclusions from these measurements are not exactly right and mislead the readers. 

Round 3

Reviewer 2 Report

Comments and Suggestions for Authors

The authors stated in their 2ns revision that “the same thickness of SiNx films and the same oxidation time of SiNx films ensure that equal amount of SiNx:O chromophore exists” (line 90 ~ 91). However, I can’t see any proof for this statement. I already asked for standard absorption spectra instead of the normalized extinction spectra as shown in Figure 3(a) [which only prove the absorption maximum instead of the amount of SiNx:O chromophore] to prove equal amounts of SiNx:O chromophore were used to compare PL and EL intensities. Where is the proof for their statement that using the same amount of SiNx:O chromophore for PL and EL intensities.
